# The Past, Present and Future Direction of Government-Supported Active Aging Initiatives in Japan: A Work in Progress

Yoshiko Someya [1] and Cullen T. Hayashida [2,*]

[1] Health & Medicine Paradigm Shift Consortium (HMPSC), Tokyo 108-0071, Japan; someya@air-r.nir.jp
[2] Department of Sociology, University of Hawaii, Honolulu, HI 96822, USA
[*] Correspondence: cullenhaya@gmail.com

**Abstract:** Active aging programs are seen as an important strategy for the long-term sustainability of Japan given population aging and fertility decline trends. This paper reviews Japan's commitment to active aging initiatives since the 1960s with a focus on the development of senior clubs, welfare centers for the elderly and senior colleges. The changing patterns of their popularity are discussed in relation to the increased options available today and the changes taking place in the family structure with both a macro historical review and a case study to demonstrate how programs have been implemented with national and local funding support. A description of the U.S. experience is used to demonstrate the comparative level of commitment that Japan has made to support healthy aging. The recrafting of the active aging motif as *shogai gen'eki*, with its emphasis on continued employment, may suggest a redirection of the preferred role of Japan's older adults in the future.

**Keywords:** Japan; active aging; senior clubs; senior colleges; lifelong learning; economic sustainability

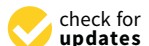



## 1. Introduction

Japan today is at an existential crossroads. As it takes the lead in becoming the world's oldest society with a strong depopulation trend going forward, all the world is watching to see how it proceeds to find a path to achieve a sustainable direction for the remainder of the 21st century and beyond. Never in its history has it found itself as a super-aging society. The size of Japan's older adult population is approaching the 30 percent threshold and still trending north (Kaneko 2008)[1]. There are significant social challenges resulting in the simultaneous need for sustainable economic development, massive debt control, disaster management, challenging international relations while promoting intergenerational integration, and a vision of sustainability for those entering preschools today. While there are initiatives at play to address its problems, poor planning could result in the freefall of this depopulation trend with the specter of societal collapse.

This paper provides an overview of the development of active aging measures initiated by both the national and local governments in Japan since the 1960s. As the world's fastest case of population aging, the documentation of Japan's path of active ageing initiatives will be closely observed by other depopulating countries trailing along this same path. The Japan government-inspired and -initiated active aging programs contributed significantly towards aging policies, programs and funding for the past 60+ years. Times have changed and now mid-course corrections are required. Indications of this need for policy change stems from population aging, urbanization, family structural changes and the political economy among other factors. The various changes are discussed further in this report.

To fully understand the rationale for this report, it is first necessary to elucidate four points of our paradigm. The first is population aging. The growth of older adults represents one of the most challenging issues for the 21st century for Japan and the world. The latter is the combined impact of declining birthrates, increasing life expectancies and the shrinking

working age cohort. The challenge is not just with having more older adults but with the increasing number of older adults who are frail, experiencing multiple chronic disabilities, and are in need of numerous long-term care and support services. In addition to becoming an enormous financial burden, it will impact the search for talent and pose defense and security challenges.

Secondly, population aging is resulting in the threat of intergenerational conflicts. The generational transfer of resources from working adults to the dependent old is building a sense of disillusionment and pessimism among young people who are paying into senior pensions, health insurance and other support services that they may not fully benefit from. The potential for long-term intergenerational conflict can weaken societal solidarity and will need to be addressed.

Thirdly, gerontological frameworks have been biased towards the medical model. During this period of rapid change, policy leaders and legislators will be in search of conceptual schemes and theoretical models to create road maps for positive social action. Public policy has become heavily dependent on gerontological models focused on aging from a sick care or the medical perspective. While it is important to support those in need of long-term care, an overemphasis on the medicalization of long-term caring and the pre-dominance of an institutional bias is unaffordable. The sustainability of society as a whole may require a reset.

Lastly, an alternative approach that this report highlights is the active aging model. There is, today, a growing interest in viewing older adults as an asset. Active aging places a greater emphasis on increasing our health span and not just on increasing our life span. There is growing research evidence that living with an active aging lifestyle can reduce the demand for long-term caring, enhance well-being, reduce healthcare costs and promote intergenerational solidarity. Conceptually, there are many related concepts such as productive aging, successful aging, healthy aging and the like. There are also considerable discussions in the literature regarding the multi-faceted nature of active aging. Active aging, for example, has been said to involve physical fitness, nutritional fitness, financial fitness, social fitness, emotional fitness, fourth age planning and purpose.

The theoretical question that can be asked is whether active aging policy, programs and funding support are viable means of reversing or moderating the economic challenges of population decline? We do not have an answer to that question yet. In the meantime, population decline is a trend affecting more than 30 countries today and is expected to affect many more in the decades ahead. Japan is at about 1.37, Korea is 0.84 and China at 1.3—all pointing to foreboding population decline. However, given that Japan is a super-aging society of the first order and given Japan's foresight in creating a large national infrastructure to promote active aging programs from the 1960s, it is imperative that an attempt be made to document what happened since the 1960s and what is now happening in Japan. This report is an attempt to identify markers of what transpired during the past 60+ years of active aging programs led by the national government.

The Japanese national government's early efforts to create an active aging infrastructure emerged in the 1960s to encourage older adults to maintain healthy, meaningful, and independent lives. Based on Espin-Anderson's perspective, Japan patterned its plans after northern European countries rather than Britain (Espin-Andersen 1990).[2] The initiatives took by the Japanese government resulted in a social welfare system quite similar to Germany's social insurance model (Someya 2016).[3]

While many of these active aging programs have been temporarily closed in early 2020 because of the COVID-19 pandemic, this paper provides an overview of the roles, functions, and interplay of three active aging programs in Japan since its inception 60 years ago—the senior clubs, welfare centers for the elderly and senior colleges. Our review will involve a macro socio-historical assessment of Japan as well as a microanalysis based on one case study. While this case study may not be representative of Japanese municipalities, it provides demonstrated evidence of the implementation of active aging funding and policies at the local level. This paper closes with a brief overview of the U. S. experience

to provide additional comparative insights regarding the future direction of active aging initiatives in Japan. The recently emerging *shogai gen'eki*[4] approach for the active aging strategy may be a precursor for Japan's evolving attempt at marshalling the power and potential of older adults for the future.

## 2. Historical Background of Active Aging in Japan

After the end of World War II, survival was the priority given the massive devastation of Japan. However, by the 1950s, the economy and life in Japan dramatically improved, resulting in many welfare and social security laws to enhance the quality of life and the legal rights of all its citizens. Among these were the mandatory medical insurance and the national pension systems in 1961 and the Welfare Law for the Elderly in 1963.[5] The 1960s was thus the decade for the fundamental change in Japan's social welfare system.

During that decade, the ideal typical family was three-generational, with an older couple living with their oldest son's family and surrounded by grandchildren. At that time, however, most retirees were still without pensions as pension was only available to the privileged few employed by government, schools, or the military. Those with that financial security had the time and opportunity to access organized recreational activities.

Within this context, the Welfare Law for the Elderly was enacted to start the universally available national pension and the medical insurance for the health and well-being of older adults. New measures began promoting healthy and active aging via social and cultural activities and volunteer opportunities. Under the 1963 Welfare Law for the Elderly[6], national funding made low-cost or free programs such as senior clubs, senior centers, and hostels for seniors (*Rojin Kyuyo Homu*) widely available (Someya 1983). The Japanese economy was now on its way towards a speedy recovery, as reflected in the inaugural start of the new bullet train from Tokyo to Osaka and the commencement of the 1964 Tokyo Olympic Games. The people of Japan were optimistically focused on enjoying their leisure time and becoming active consumers.[7]

Japan's traditional multi-generational '*ie*' family system, which was strengthened during the modern era, was evolving into a nuclear family structure. With this change, the position and influence of older adults in the family declined. Older adults were beginning to feel disregarded, lonely and started to seek social relations outside the family. In 1960, 86.8% of those 65 and over lived within a three-generation family arrangement. Since then, the proportion of three generation families has continuously declined such that by 2018, only 32.5% of Japan's families still maintained the traditional family pattern (Someya 2003).[8] By contrast, older adults living alone grew from 3.8% in 1960 to 27.4% by 2018.[9] Today, the most common living arrangement among older adults is living in spouses-only households (32.3%). However, when one of the spouses passes away, many older adults remain alone rather than with their adult children.[10]

By the 1980s, the Japanese economy and the pension systems had matured, and older adults became increasingly independent to pursue self-enrichment activities. Their loss of status as heads of multi-generational households was replaced with increased financial independence creating a new generation of seniors enjoying a plethora of leisure activities. This change is the basis for our attempt to understand the underlying basis for the development of active aging programs in Japan. What follows is a description of select programs at the national level and a case study of their implementation in Urayasu-city on the outskirts of Tokyo.

## 3. Development of National Active Aging Programs

Three types of active aging senior programs will be discussed here. These are the senior clubs, the welfare centers for the elderly, and senior colleges. While there are many other senior programs, such as the Silver Human Resource Centers and Elder Hostels, these three are essential to understanding the nature, character, and direction of active aging-oriented programs in Japan. While the term "active aging" is recognized among Japanese gerontologists, it is not as widely known in comparison to terms such as *ikigai (life*

*purpose)* in Japanese literature and popular culture. Active aging as a theoretical construct is still evolving but has generally emphasized the multi-faceted aspects of health, wellness, social engagement and productivity Do not delete "y"(Walker 2008).[11] To that extent, there is correspondence to the Japanese concept of *ikigai*, although this latter term seems to place a higher premium on to meaning, purpose and self-actualization by comparison (Kumano 2018).[12]

**Senior Clubs (Rojin Kurabu)**

Senior clubs are found in communities throughout Japan. These are informal community-based social groups that local governments financially support. Community ties were strong in the 1950s and 1960s, and most older adults were actively involved within their communities. With government support, senior clubs in neighborhoods evolved from their informal social pattern to become increasingly popular and assumed a fundamental role in enhancing active aging. Records indicate that the first senior club was formed in Yokaichi-city in Chiba Prefecture in 1946.[13] In 1952, the Social Welfare Council promoted the launching of senior clubs throughout Japan. In 1962, the Japan Federation of Senior Citizen's Clubs was established, and the national government supported their expansion with subsidized funding.

In 2020, there were 92,836 senior clubs with about 5 million persons over 60 years of age registered.[14] While they still represent the most popular form of active aging program today, participation rate has trended downwards.

The peak of their popularity was in the 1980s, when 48% of all eligible seniors registered with a club. By 2015, the national participation rate among older adults had dropped to 15%. However, there is considerable variance in the participation rate from 47% in Toyama Prefecture to only 5% in Kanagawa Prefecture adjacent to Tokyo.[15] Figure 1 depicts the rise and decline in the national participation rate of senior clubs. Financial independence, higher education, improvements in the Japanese economy, and the universal pension system have provided more options though not necessarily less interest in social participation among older adults. Modernization and urbanization have also made older people more individually oriented. As a result, many may prefer to join non-age segregated hobby groups and cultural circles, including those organized by for-profit organizations[16].

Nevertheless, while competing with newly emerging opportunities, senior clubs continue to be quite popular, especially in depopulated areas where social clubs organized by for-profit organizations are less likely to exist. Each club plans indoor and outdoor programs according to its members' preferences. Based on a senior club survey conducted by Japan Federation of Senior Citizens' Clubs Inc. in 2015[17], the average club member was 67 years old. Over 40% were 80 years of age or older. Approximately 60% were women.

Once a club has at least 30 members aged 60 or over, it can receive a subsidy from its municipal government. In 2020, the annual national assistance provided to each club amounted to approximately USD 465 for those with at least 30 members. The total allocation for senior clubs from the national government in 2021 was USD 25,719,350. Given that both the prefectural and local municipal governments match that contribution, the total support amounts to triple that amount.[18] In addition, about 63 percent of the clubs surveyed indicated that they required about USD 15 as their annual membership fee.

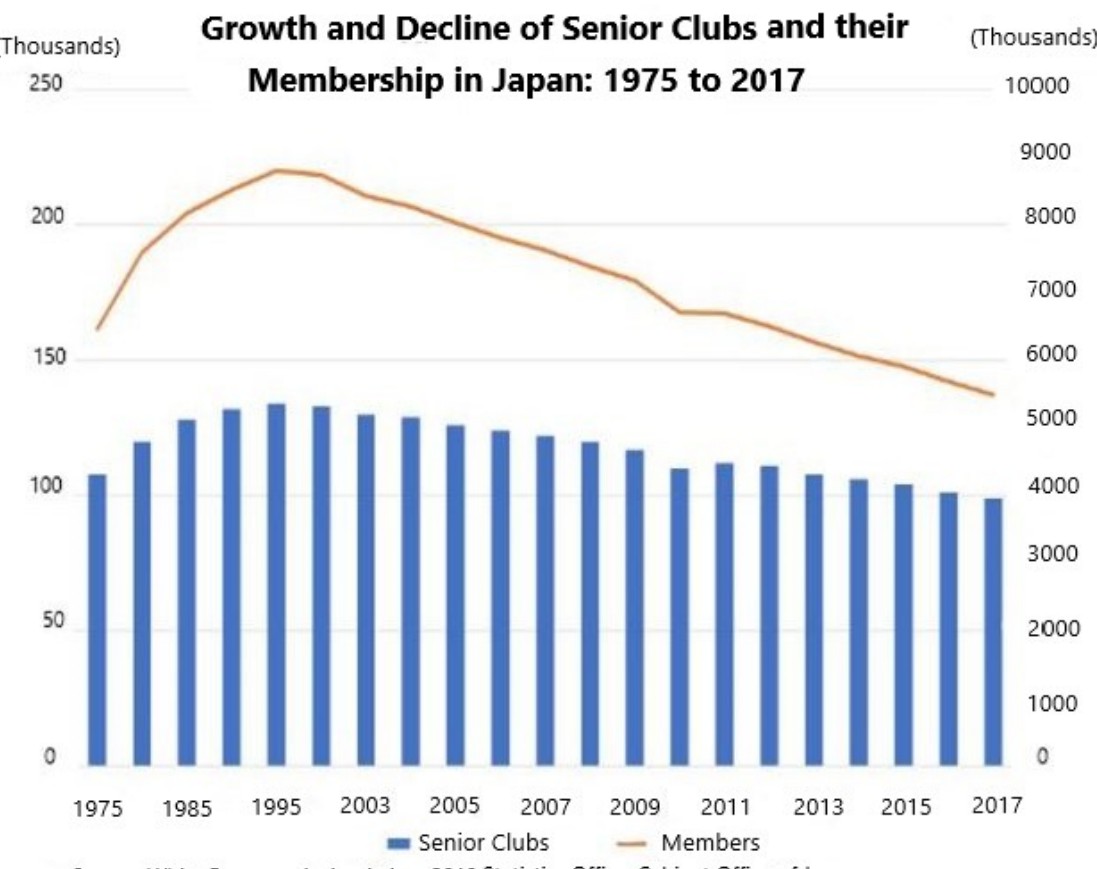

**Figure 1.** Growth and decline of senior clubs and their membership in Japan—1975 to 2017.

Reported group activities ranged from health care, social friendship activities, charity fund-raisers, volunteer activities, educational and hobbies activities, community service projects, production of local products, and oral history sharing. As noted in Table 1, there is a high degree of similar activities among reporting clubs with educational hobbies, volunteering, and health as the most common. More specifically, a survey of 1374 clubs noted that the physical–social activity called ground golf was the most popular among 150 clubs.[19] Others that were also popular included the cleaning of public places, hobby circles, annual meetings, educational excursions, karaoke singing, and other sports-related exercise programs. While less frequently mentioned, other groups engaged in friendship-exchange programs, health promotion lectures, birthday parties, participation in community events, assisting children at crossings, and engaging in the beautification of public green spaces.

**Table 1.** Predominant types of activities by senior clubs.

| Type of Activity | Percent Offering Activity | Average Number of Programs per Club |
|---|---|---|
| Educational/Hobbies | 95.70% | 3.2 |
| Volunteer | 94.80% | 2.7 |
| Health care | 92.50% | 3.4 |
| Community Service | 91.70% | 3.5 |
| Social/Friendship | 84.00% | 2.4 |
| Making Local Products | 49.10% | 0.9 |
| History Sharing | 49.10% | 0.9 |

Source: Japan Federation of Senior Citizens' Clubs Inc. (2015). *National Survey Report on Senior Clubs in 2015.*

Senior club activities can also be understood by grouping them into three categories. The first is indoor hobbies such as karaoke, crafts, painting, Japanese chess, and educational meetings. The second includes exercise programs such as ground golf, gate ball, Chinese qigong, yoga, and stretching exercises. Additionally, the third cluster of programs involves volunteer and community service projects, including but not limited to cleaning parks and streets, assisting school children at street crossings, and friendly visitations of the home-bound. Collectively, these activities help older adults maintain their physical and mental health, remain socially connected, and feel helpful and of service to their communities filling their sense of *ikigai* (*life purpose*).

**Welfare Centers for the Elderly (*Rojin Fukushi Senta*)**

Welfare centers for the elderly (*Rojin Fukushi Senta*) are community centers for retirees developed by the quasi-governmental Social Welfare Councils (*Shakai Fukushi Kyogikai*) at the local level. Starting in 1974 under the Social Welfare Law for the Elderly, welfare centers, as physical facilities with paid staff, function as community gathering places for older people at little or no cost to them. Older adults living in the community are eligible to be members. These centers have played a key role in providing active aging services and programs in communities to the present.

Officially, there are three types of centers. Type A facilities provide social service consultations (e.g., life issues, housing, relationships), health consultations, health promotions (nutrition, exercise), employment and vocational workshops, rehabilitation, education, and support to senior clubs. Another is referred to as Special Type A facilities, which provide more health care services with clinics for health examinations, health counseling, nutritional guidance, rehabilitation services and exercise. Special Type A facilities are also available for recreation and the use of its health reference rooms. Some also have a public bathhouse (*sento* or *onsen*) on-site as well. The third, known as Type B facilities, are smaller in size and complement the larger centrally located Type A facilities in neighborhoods. They also provide consultations related to housing, social relations, health issues, lifelong learning opportunities, and support to senior citizen clubs in the area. Collectively, these facilities provide programs for older adults to maintain their physical, mental, social, and financial well-being. As of October 2017, there were 1376 Type-A facilities, 246 Special Type A facilities, and 444 Type B facilities in Japan.[20]

While the number of these government-funded facilities has remained stable, informal reports suggest that while these welfare centers for the elderly continue to provide many services, participation has declined in some areas[21]. For example, at the Urayasu welfare center for the elderly, the percentage of participants to the eligible population dropped from 45.6% in 1974 to 25.8% in 1980, 17.9% in 1990, 10.0% in 2000, and 8.3% in 2004, respectively.[22] This seems to be a pattern similar to what has been happening to senior clubs in Japan.[23]

**Senior Colleges (*Koreisha Daigaku*)**

Since the 1970s, Japan has witnessed a growing number of silver or senior colleges. Fuwa points out that the history of contemporary adult or lifelong education in Japan stretches back to 1949, when the Act for Adult Education was enacted (Fuwa 2001).[24] By 2002, adult education centers or programs were established throughout Japan in over 18,000 community centers (*kominkan*) and have used the Japanese word for college or university loosely for them (Ogden 2010).[25] Today, they continue to provide an essential role in community education. In addition to community centers, universities also offer a variety of non-credit educational courses. Entire programs for older adults may range from community lecture series to those that are two years in length and culminate in a completion certificate. Course offerings cover a variety of topics to enhance interest in lifelong learning. While it is difficult to follow the exact path of policy development, lifelong education as a concept was formally adopted by Japan following the 1972 UNESCO's Faure Report (Faure et al. 1972).[26] From 1973, local governments began encouraging educational programs for older adults, and a limited number of initiatives emerged.[27] In 1984, the Integrated

"*Ikigai*" Promotion Program was enacted, and in 1989, "The *Choju Gakuen* (Long-life School)" was established.

By 1990, the Master Plan for Lifelong Education and the Law for the Promotion of Lifelong Learning was established. This law created a Lifelong Learning Council at the national level and the prefectural level. It facilitated the establishment of senior colleges in large cities with over 1 million residents and programs in each prefecture. Within the next decade, the Ministry of Education, Culture, Sports, Science, and Technology (MEXT) created its Guidelines for the Development of Lifelong Education. These government initiatives resulted in the emergence of silver or senior colleges in the 1980s, with approximately 50% of all prefectures and cities with at least 1 million residents operating senior college programs (Takahashi 1999).[28] Kinoshita points out that programs are offered by local municipal governments, voluntary learning groups and in recent years, by urban universities (Kinoshita 2018).[29] Setagaya Ward senior college (*Setagaya-ku Roujin Daigaku*) in Tokyo was an early well-known program (Miura 1996).[30] The Setagaya Ward senior college, organized by its municipal government, is in one of the most populated residential parts of Tokyo, where most of its residents are white-collar professionals and college educated. Examples of other noteworthy senior college programs can be found in Rikkyo University's Second Stage College Program in Toshima Ward of Tokyo, created in 2008 (Kinoshita 2018).[31] The Rikkyo program has three-course clusters: the Aging Society Education, Community Design and Business, and the Second Stage Design. Others, such as the *Doubou Gakuen* in Kitakyushu city, the Silver College of Chiba Prefecture, Nakano Long-life University, and the Sendai Good Age Academy in northern Japan, are also well-regarded examples.[32] Improvements in Japan's living standards and lifestyles of older adults have increased opportunities to engage in lifelong learning through emerging senior colleges.

## 4. A Case Study of Urayasu City

Urayasu-city, located on the eastern outskirts of Tokyo, provides a useful case-study of these two active aging-related programs. The city is the popular bed-town for Tokyo workers and is better known for Tokyo Disneyland. The city was established in 1981 with a population of about 70,000 and grew over the next 40 years to more than 170,000 residents by 2020. Those over 65 represent 17.4% of the population. While Japan has an older adult population 65 and older, representing 28.7% in 2020, Urayasu city, by comparison, is one of the younger cities. Urayasu city has an excellent financial base with a significant proportion of upper-middle-income residents and an average annual income that ranks 18th among 1741 municipalities in Japan.[33] The financial status of Urayasu is reported to be the best among all local municipalities in Japan.[34] Consequently, Urayasu city has been able to provide a significant amount of financial support for community and older adult services.

As a case study, Urayasu may be considered too atypical to base generalizations of Japan's active aging policy, practices and trends upon. Nevertheless, this area was selected because of relational opportunities for multiple interviews to access detailed policies and practices. While many municipalities do not have the financial confidence nor capacity to implement national policies as well, our conjecture is that other cities would probably pursue a similar direction if they had the means. The following measures reflect examples of its financial confidence, capacity and commitment.[35]

First, there are 50 senior clubs throughout Urayasu-city, with one in each neighborhood. All but seven have their own 'clubhouse' built with city funding, and plans are underway to ensure that the remaining clubs will eventually have their place. Aside from 2020, when the COVID-19 pandemic hit and active aging programs in Urayasu city were temporarily closed, financial support for active aging programs has been substantial. In 2019, for example, senior clubs in Urayasu were allocated USD 73,000 from the city and another USD 12,000 from the national government for their programs. In addition, they received an allocation of USD 23,000 for utility expenses and over USD 11,000 for maintenance and repairs for club houses.

The year 2019 was also when the city financed the construction and operations of two new clubhouses for over USD 870,000. In addition, Urayasu Association of Senior Clubs which coordinates and supports all Urayasu senior clubs was allocated USD 45,000 by the city.[36] Additional financial support was also obtained from the Urayasu Social Welfare Council.

Second, Urayasu Welfare Center for the Elderly facility, called the "U Center", is significantly financed by the Urayasu Social Welfare Council. The U Center provides numerous cultural and leisure activities, rehabilitation, and counseling programs. In addition, the center has a coffee lounge and a public hot bath (*sento*) for its senior residents. The city government funds the center by way of the Social Welfare Council. In 2020, USD 1,482,000 was allocated by the city to cover the facility's operational cost. In addition, activity leaders and administrative staff are also financed with city funds.

Third, the Silver Human Resource Center is another example of an active aging infrastructure found throughout Japan funded with the Urayasu-city and the national government funds. These centers are senior employment and placement agencies that help introduce job openings to seniors. Many of these job openings are below minimum wage, low-skilled and part-time assignments involving gardening, bicycle repair, parking assistance, assistance to children at street crossings, and the like. Non-profits operate these senior employment agencies and help supplement the pensions of older adults.

Fourth, the Urayasu Citizens' College, which started in 2008, received USD 180,000 in 2019 in support of its educational courses and open lecture program. A more detailed discussion about senior colleges in Urayasu follows in the section below.

Fifth, transportation services represent another infrastructural and financial support for the active and frail older adults in Urayasu. In addition to the well-developed train system linking the city to the rest of Japan, the local bus lines also crisscross the entire city. There are also community mini-buses covering less travelled routes that cost is USD 1 per ride and otherwise free for elementary school children and toddlers. Older adults over 70 and the disabled are provided with a voucher worth USD 56 of bus tickets annually regardless of income. Over USD 760,000 was budgeted in 2019 to support these free bus rides for seniors and disabled people. In addition, another USD 100,000 is budgeted to cover the cost of four large buses capable of accommodating wheelchairs specifically for senior citizens and all citizen groups to use for group excursions and transportations for events.

Finally, Urayasu city also budgeted USD 490,000 in 2019 for celebratory gifts to older adults upon the arrival of the 77th, 88th and 99th birthdays given the special cultural meaning that the attainment of these ages means to the Japanese public.

**Urayasu Senior Clubs**

The first Urayasu senior club was established far earlier than most in April 1962.[37] By 2020, there were 50 clubs for Urayasu's 37,356 older adults 60 and over. These 50 clubs had 3356 registered club members representing 8.9% of all older adults. Based on 2015 national data, Urayasu's participation rate was markedly lower than the national rate of 15%.[38] Each club receives an annual subsidy depending on the size of its membership. Clubs with fewer than 50 members receive USD 1200 annually. Contributions increase to USD 1500 annually for clubs with 50 to 100 members and are capped at USD 1900 for memberships over 100. These funds are from the national government that channels it through the prefectural and municipal governments. As of 2021, 43 of the 50 senior clubs of Urayasu had club houses with centralized heating and solar panels built at city government expense. In addition, the club's monthly utility fee, semi-annual deep cleaning, and disinfecting services are underwritten with city funds. In addition to these subsidies, individual clubs often require a nominal membership fee. For example, the Chowa-kai club in Urayasu requires a monthly membership dues of USD 1.

The 50 Urayasu senior clubs are all part of the Association of Urayasu Senior Clubs. That organization has been instrumental in supporting the implementation of programs to promote health, senior sports, hobbies, culture, lifelong learning, recreation, and a variety of

training workshops. The association has also supported programs that encourage members to contribute to their communities. This includes friendly visiting, volunteering, storytelling, intergenerational activities, gardening, handicraft activities, community cleanups, and recycling activities.

While the COVID-19 pandemic has temporarily suspended club activities until late 2021, Table 2 displays the array of activities that one senior club (*Chowa-kai*) in Urayasu city plans on a monthly and weekly basis for an entire year. The *Chowa-kai* senior club started in 2007, and in 2021, it had 13 men and 30 women in the club. With a total of 43 members, the club has an annual budget of about USD 2266. The city provides additional in-kind financial support for meeting space, staff support, and the free use of the city's sightseeing bus twice a year for day excursions. Lunches and admission fees for day tours are born by the participants.

**Table 2.** Monthly and weekly schedule of the Chowa-kai Senior Club of Urayasu-City.

| Monthly Schedule | Planned Activities |
|---|---|
| APRIL | • General meeting of Urayasu senior clubs<br>• Annual meeting of Chowa-kai |
| MAY | • Visit to a local elementary school and kindergarten<br>• Personal safety check for the members |
| JUNE | • Spring one-day educational excursion<br>• Summer community cleanup volunteering |
| JULY | • Lecture by a guest speaker<br>• 1st workshop for club leaders |
| AUGUST | • Workshop for club members—health checkups<br>• 2nd workshop for club members |
| SEPTEMBER | • Annual performing art for senior citizens<br>• Golden wedding anniversary celebration for members<br>• 3rd workshop for club members<br>• Lecture by a guest speaker |
| OCTOBER | • Field trip to Irifune Kindergarten<br>• Autumn fair at a condominium<br>• 4th workshop of club members<br>• Autumn one-day tour |
| NOVEMBER | • Autumn community outdoor cleaning volunteer work<br>• 5th workshop of club members |
| DECEMBER | • Exchange party at a local kindergarten<br>• Workshop on community health<br>• Year-End party |
| JANUARY | • New Year party for club leaders—Urayasu Senior Club Association |
| FEBRUARY | • New Year party<br>• 6th workshop of club members<br>• Annual General Meeting of Urayasu Senior Clubs |
| MARCH | • 7th workshop of supporting club members<br>• 8th workshop of supporting club members |
| **Weekly Schedule** | **Planned Activities** |
| DAILY | • Morning exercises—6:30 a.m.–6:40 a.m. |
| 1st MONDAYS | • Monthly meetings of all senior club leaders of Urayasu City |
| 2nd THURSDAYS | • Chowa-kai club staff meetings |
| 3th SUNDAY | • Natural dsaster prevention meetings<br>• Club meeting at the condominium |
| WEEKLY | • Mahjong club activities |

Source: Annual Report of Chowa-kai Senior Club.

**Urayasu Citizen College (Urayasu Senior College)**

The senior college of Urayasu-city was established in 2008 by the city. The purpose of lifelong learning by this program is to encourage the joy of learning and to create opportunities for older adults to enhance the community's well-being. Education focuses on building stronger communities by utilizing acquired knowledge and skills through the various courses provided.[39] The three categories of senior college learning are Social Relations and Networking (*Deai* classes), Awareness of Urayasu City issues (*Kizuki* classes), and Engagement with City Development activities (*Ninau* classes). Participants can acquire skills to link the municipal government, private industries, non-profit organizations, and community residents. The 2020 schedule of the three types of senior college courses of Urayasu city is displayed in Table 3. *Deai* classes provided opportunities to meet others and to encounter new experiences. These classes promote new social relations and increase understanding of healthy brains and bodies, history and future of Urayasu, the role and function of government, population aging, law, and the economy. *Kizuki* classes focused on more specific topics such as the health and medical care system, finding a new career in retirement, reviewing city plans for the next generation, companion pets and pet care in the city, environmental issues, and awareness of cultural differences. Finally, the *Ninau* classes focused on challenging students to engage by editing a new geographic publication about the city, beautifying the city by planting flowers and growing a garden, preparing to mitigate natural disasters, assisting frail older adults and addressing issues related to condominium life and its management.

**Table 3.** Examples of courses offered by the Urayasu senior college program.

| Deai Courses | Social Relations and Social Networking | Class Sessions |
|---|---|---|
| | ● Encountering the world: We are with you! | 20 |
| | ● Health culture of Urayasu city: Enjoying active life in brain and body | 17 |
| | ● Understanding the present from learning history by choosing a book individually | 10 |
| | ● The present and the future of Urayasu city; Considering the future of Urayasu city based on a good understanding of current measurements | 10 |
| | ● Learning about local governments | 14 |
| | ● Life-long education and community developments | 12 |
| | ● Economy and laws in the population aging society | 10 |
| 1-3 Kizuna Courses | Awareness of Urayasu-City Issues | |
| | ● Considering medical care and public health | 12 |
| | ● Looking into individual career life in Urayasu city (child care is available) | 14 |
| | ● Considering renewing city plan of Urayasu for the next generation | 16 |
| | ● Thinking of living together with pets: knowing more about pets | 13 |
| | ● Observing, considering, and reforming environments in own community | 8 |
| | ● Learning about different cultures | 14 |
| Ninau Courses | Engagement in City Development Activities | |
| | ● Editing the New Geographical book of Urayasu city | 20 |
| | ● Making the Garden city and planting flowers in communities | 20 |
| | ● Making own life and enjoy living in Urayasu city | 18 |
| | ● Preventing the citizen and city from the damages of natural disasters | 18 |
| | ● Preventing elderly people from the frailty in Urayasu city | 14 |
| | ● Considering the future of condominium life and its management | 12 |

Source: Urayasu Citizen College. *Syllabus of Urayasu Citizen College 2020.* (Urayasu Citizen College 2019a).

Instructors were specialists, university professors, or retired professors in each field. In 2019, class sizes ranged from 4 to 56 attendees, with a total of 261 registered. The average

age of attendees was 70.6 years.[40] Each course is either six or twelve months in length. Older adults are permitted to participate for a total of 5 years.

Admission starts on April 1. The admission fee for one 90 min class session is USD 5. If one course has ten sessions, the course fee is USD 50. Since the financial status of Urayasu city is good, the city provided more financial funding for these types of community services in comparison to other municipal governments. Thus, while Urayasu does not represent a typical city in Japan, other cities provide as much support as possible with added assistance from the national government.

## 5. Reflections from the U.S. and Hawaii Experience

In comparison to Japanese national initiatives supporting an active aging lifestyle, the U.S. (and Hawaii as a microcosm) has some common but less organized development threads. The use of this comparison reveals some commonalities with senior club and senior college formations but with noticeable variations based on their respective levels of government support. Nationally, the primary impetus of active aging governmental initiatives in the United States may be said to have emerged via the Corporation for National and Community Service, an independent agency of the U.S. government. This agency oversees a variety of community service programs for youth and older adults. Under its older adult initiative known as Senior Corps, three programs support volunteer activities for those 65 and older. The first is the Foster Grandparent Program, which recruits older adults to teach and mentor children in elementary classroom settings. The second, known as the Senior Companion program, recruits low-income older adults to serve as companions to frail older adults to promote independent living and aging in-place in their own homes. Low-income Senior Companions volunteers are provided a modest stipend (below minimum wage) for their services. Additionally, the third program, known as RSVP (Retired Senior Volunteer Program), is more flexible in how and where they dispatch senior volunteer assistance. In many instances, the RSVP senior volunteers are persons 55 and older, providing critical community services and high-quality experience enriching the lives of volunteers. The Senior Corp programs were formed around 1993 and are available throughout the United States.[41]

Senior centers are also another important program supporting the active aging lifestyle. However, support for these programs varies widely, with county and state funding support unevenly distributed from region to region. Many are also highly dependent on foundation grants and membership fees to cover their operating costs. While the non-profit organization called the National Council on Aging maintains a National Institute of Senior Centers, it has primarily played a coordinating role for educational meetings and the establishment of national standards for center operations. It is uncertain, however, whether the credentials that are offered are widely recognized or sought.

Finally, the U.S. government created the Older Americans Act in 1965, resulting in national funding for older adults and the National Administration on Aging, the State Units on Aging for all 50 states, and the Area Agencies on Aging at the county or local level. This national initiative has supported nutrition and home and community-based supportive services, health promotion, and the national family caregiver support program. While the range of services this Act supports is wide-ranging, Section 5 of this Act explicitly supports low-income seniors in community service employment and volunteer opportunities in coordination with the aforementioned national Senior Corps program. As far as senior college initiatives are concerned, no known national government initiatives are taking place in the U.S. today. However, the non-profit Road Scholar Program (formerly known as Elderhostel) is one of the largest educationally oriented travel programs in the world targeting older adults. It has provided educational journeys to over 100 countries, and in 1999, it enrolled over 250,000 lifelong learners.[42]

**Hawaii's Programs**

The national Senior Corp and the Administration on Aging initiatives are a part of the Hawaii eldercare ecosystem. Hawaii has several senior center facilities, but their ability to access government funding is uneven. Two that have been successful are the Lanakila Multipurpose Senior Center and the Moiliili Senior Center in Honolulu. Both are non-profit organizations that operate their programs with financial support from the State government. In addition, senior centers on the other island counties of Hawaii receive support from their county governments. In contrast, other senior centers on Oahu are dependent on grants, fundraising activities and membership fees.

The senior clubs in each county in the State of Hawaii operate independently of the other counties, and there is no overarching organization coordinating, linking or supporting their collective mission. These clubs are generally organized informally and can use community center facilities for their meetings at no cost. Coordination with their respective county offices on aging varies considerably given the State's Office on Aging lack of direction.

Approximately ten years ago, as many as 55 senior clubs registered with the City and County of Honolulu's (Oahu Island) Parks and Recreation Department. Today, the Department provides physical space and limited staff support for about 35 clubs serving approximately 3000 retirees.[43] The drop in their popularity may be related to the availability of other senior club opportunities provided by churches and non-profit organizations.[44] In addition, compared to the earlier generation of retirees during plantation era in Hawaii, retiring older adults today are less aligned with their geographic community or ethnic groups when seeking social relations.

As noted above, senior clubs registered with the city government operate without funding support. They independently organize various educational, social, recreational, physical exercise, and field trip activities nine months of the year except during the summer months when the community centers are used for youth programs. Often, these clubs will assess nominal annual dues such as USD 10 per year that the club itself manages entirely. Participation in these clubs is open to any senior wishing to participate. While these clubs have not been permitted to physically meet at municipal park facilities during the COVID-19 pandemic, those who were familiar with computer use adjusted to virtual meetings as a means to socialize and receive current updates such as vaccinations.

For mostly older men who were not interested in club activities, the City of Honolulu's Parks Department supports the 50 and older sports-minded individuals among them with its the Senior Slow-Pitch Softball League that began in 1975 on Oahu. There are about 20 teams on Oahu and comparable leagues for each of the other island counties. This program provides another opportunity for physically active older adults of all ethnicities to strengthen social relations, engage in physical activity, and have fun.[45]

In Hawaii, educational opportunities for older adults have been expanding. The University of Hawaii's main campus in Honolulu has a Senior Citizen Visitor (*Na Kupuna*) Program, which offers access to many college courses for free for those 60 years of age or older without university credit. Older adults must register after the regular enrollment period but before classes start with the university's program office. They are also required to complete any required health tests and obtain written permission from the course instructor. Aside from graduate and professional degree courses programs, retirees are permitted to register for up to two classes per semester. A comparable program is also available through each of the community colleges throughout the State of Hawaii.[46]

Another program at the University of Hawaii's main campus is the Osher Lifelong Learning Institute. This is nationally affiliated with the Bernard Osher Foundation that has financially supported 120 lifelong learning centers throughout the U.S. since 2001 for those 50 and over. The University of Hawaii's Osher Institute provides non-credit courses, workshops, lectures, special events, film series, and other activities to encourage older adults to engage their minds, enrich their lives and serve the community. Membership is

USD 60 per semester and that entitles members to enroll in up to three specially created non-credit courses and three special events during that time. In addition, membership provides university library privileges, discounts at the university bookstore, and rides on the university's shuttle bus service. Presently, this program has over 500 registered members participating each semester.[47]

## 6. Summary and Conclusions

In Japan, active aging measures have developed as a national government initiative since the 1960s. During the past half-century, the Japanese government initiated various active aging efforts that have greatly contributed to healthy ageing and possibly extending the nation's life expectancy. Since the 1960s, urbanization trends, economic growth, high literacy rates (Saito 2012 )[48], and changes in the living arrangement from living with adult children's family in old age to living only with one's spouse or alone have buttressed the support of national active aging initiatives.

This study principally reported on two programmatic trends in Japan, one with senior clubs/senior centers and the other with senior colleges. With senior clubs and centers a series of national policy directives and funding contributed significantly to the rapid growth of these programs, facilities, and staffing nationwide. There has been evidence of a decline in the number of clubs and their level of participation over time. Nonetheless, their role in supporting older Japanese to remain active and engaged with various community-based programs remains strong.

While there has been evidence of a decline for senior clubs and their level of participation, senior colleges, which have also received some policy and financial support, may be seen as another avenue for active aging program development.[49] Municipalities and universities have initiated them. While reliable statistics are not yet available, this appears to be an emerging form of active aging worth monitoring. Program participation rates appear far lower than senior clubs. Together with the COVID-19 pandemic since 2020, mitigating factors may be their cost and their time commitment as older adults continue working. Will it replace the role of senior centers and senior clubs? Probably not. However, urbanization, increased life expectancy, economic well-being among older adults, and their high literacy rate will continue to create more options and choices for older Japanese today.

By comparison, the role of government at the national, state, and local levels in the U.S. for active aging is significantly less deliberate. The active aging lifestyle's support from the government is uneven and more dependent on non-profits, eleemosynary foundations, and membership fees. County government support of senior clubs varies by location and coordination seems limited. In addition, there has been a reported decline in senior clubs on Oahu Island managed by the county government. Like in Japan, this may be due to the increased number of options and opportunities available to older adults. Seniors now have more choices.

Likewise, senior colleges in Hawaii have had a similar program to Japan's. However, it has neither been officially supported at the state nor at the national level by policy or funding. Programs such as the Osher Lifelong Learning Institute at the University of Hawaii are affiliated with a nationwide network of similar programs throughout the U.S. but primarily supported by a private foundation. The national Road Scholar travel educational program likewise does not have a visible presence in Hawaii. Active aging measures are less embedded into the social fabric of American society and into the eldercare social infrastructure. However, educational programs for seniors and adult learners are expanding. This is not due to government interventions and initiatives yet but due to increased offerings via the internet.[50]

For Japan and the U.S. as a whole and via Urayasu-city and Honolulu as microcosms, changes have supported active aging-related initiatives. Japan has provided solid financial and policy support at the national and local levels, whereas this has not been reflected to the same degree in the U.S. With economic change, increases in education, urbanization,

and other trends, older adults in both countries are being provided with more choices and options. This trend will probably continue for the foreseeable future.

Is the active aging strategy effective? This is a complex question since, as noted earlier, active aging is multi-faceted and it is still uncertain which of its facets is most important. This paper cannot answer to that question yet. Nevertheless, early indications suggest that the active aging strategy will not be abandoned but will continue to evolve. Changes in the family structure and urbanization have altered options available to older adults. When the authors reviewed the conceptual discussions of active aging as a newly emerging field in international gerontology, we found that active aging seems to have started with a primary emphasis on physical fitness and self-enrichment. The development of senior clubs and senior centers are in keeping with that approach in Japan. Another international active aging literature is forming around lifelong learning. This is reflected by the emergence of the senior college programs in Japan, Hawaii and elsewhere. We have also observed active aging with a focus on developing purpose beyond oneself and with continued employment. Active aging for employability may be the direction that will be Japan's active aging 2.0 approach and direction for the latter half of this decade. In the meantime, it is imperative that what happens in Japan be documented and monitored since its implications for other societies undergoing demographic decline are significant.

With population decline, population aging, and declining fertility rates, Japan's financial challenge to maintain its level of support for past older adult programs may face future challenges in maintaining its current level of support. Tax increases are already occurring and retirement age for pensions are changing. Will Japan tamper down its generous support of older adults such as its celebratory gifts for centenarians? Will Japan need to expand its use of older adults as a human resource by expanding the role of Silver Human Resource Centers? More, however, may be needed. In concert with the government, some gerontological professional consortiums are creating a new push to promote the *shogai gen'eki* message to encourage older adults to remain employed.[51] In keeping with this government effort to encourage the continued employment of older adults, the mandatory retirement age in Japan is about to be extended beginning in 2023.[52] Perhaps active aging in Japan is shifting to expand the productive role of retiring older adults beyond senior employment centers with the approaching 100-year life. In the coming decades, both countries will need to address ways for active agers to transform from merely maintaining physical well-being and finding self-enrichment to remaining useful, to matter, and to make a difference for a sustainable society.

What will happen if the active aging approach to address societal sustainability is unsuccessful? What then? This too is only subject to speculation. We do not know how effective active aging will be as a national strategy to address the long-term sustainability of a society. It is probably important but could combine with other methods for a broader impact. From that broader perspective, Hayashida has identified other strategies beside the active aging approach to address population decline and long-term sustainability. Among these are the use of technology, the importation of foreign workers, increasing birthrates, postponement of the retirement age, age-friendly community development, deurbanization, increasing women in the workforce, and the expansion of childcare support infrastructure (Hayashida 2020).[53] To date, however, there has been no coordinated effort at monitoring and orchestrating these various strategies in Japan or elsewhere yet. This is all the more reason why future coordinated research and monitoring may be needed from all countries experiencing population aging.

**Author Contributions:** Y.S. and C.T.H. contributed equally to the manuscript. All authors have read and agreed to the published version of the manuscript.

**Funding:** This research received no external funding.

**Institutional Review Board Statement:** The study was conducted in accordance with the Declaration of Helsinki. The authors have been informed that the study does not require an Institutional

Review Board review or approval since it is based solely on secondary data available to the public. Communication with Chair of the IRB was on 31 January 2022.

**Informed Consent Statement:** Not applicable.

**Data Availability Statement:** All data are deemed information from the internet, secondary sources or from public official documents. No information was directly solicited from human subjects.

**Conflicts of Interest:** The authors declare no conflict of interest.

## Notes

1    Kaneko, Ryuichi. 2008. Statistical Foundations of Population Projections. In *The Demographic Challenge: A Handbook about Japan*. Eidted by Florian Coulmas, Harald Conrad, Annette Schad-Seifert and Gabriele Vogt. Leiden: Brill, pp. 50–53. (Kaneko 2008).

2    Espin-Andersen, Gosta. 1990. *The Three World of Welfare Capitalism*. Cambridge: Policy Press. (Espin-Andersen 1990).

3    Someya, Yoshiko. 2016. Changing Relationships between the Elderly and Their Adult Children. *Japanese Journal of Family Sociology* 28: 63–72. (Someya 2016).

4    Shogai gen'eki ("Active long life" is a slogan that is being popularized by the national government and the Committee for the Establishment and Promotion of "Lifetime Active Day" and established October 1st as "Shogai Gen'eki Day". There is strong emphasis on continued employment in retirement in its messages. Available online: www.lifelongsociety.org (accessed on 21 August 2021).

5    Welfare Law for the Elderly, (*Roujin Fukushi Ho*, in Japanese), 1963. National Health Care Insurance (*Kokumin Kenko Hoken*) in 1961, and Basic Pension (Mandatory *Kiso Nenkin*) in 1961.

6    ibid.

7    There was a strong consumer revolution occurring from 1955 to 1964. Color TV sets, home air conditioners, and cars were in high demand at that time. "White Paper on Economics" (1959). Cabinet Office of Japan.

8    Someya, Yoshiko. 2003. Changing Relationships between the Elderly and Their Adults Children. *Japanese Journal of Family Sociology* 14: 105–114. (Someya 2003).

9    Someya, Yoshiko. 2003. Changing Relationships between the Elderly and Their Adults Children. *Japanese Journal of Family Sociology* 14: 105–114; See also White Paper on Aging Society 2020 (Bureau of Statistics, Cabinet Office of Japan 2021).

10   White Paper on the Aging Society 2020, Table 1-1-8. In 2000, a 27.1% of the elderly live with a spouse and 19.7% are living alone. In 2019, living with a spouse increased to 32.3% while living along increased markedly to 28.8%.

11   Walker, Alan. 2008. Commentary: The Emergence and Application of Active Aging in Europe. *Journal of Aging and Social Policy* 21: 75–93. (Walker 2008).

12   Kumano, Michiko. On the Concept of Well-Being in Japan: Feeling Shiawase as Hedonic Well-being and Feeling Ikigai as Eudaimonic Well-Being. *Applied Research in Quality of Life* 13: 419–33. (Kumano 2018).

13   This city was renamed Sosa-city after merging with another town in 2006. Source: Japan Federation of Senior Citizens' Clubs. 2015. *The National Survey Report on Senior Clubs in Japan*. (Japan Federation of Senior Citizens' Clubs Inc. 2015).

14   Eligibility for members is defined by the platform of Japan Federation of Senior Citizens'. Japan Federation of Senior Citizens'. 2021. *Leaders' Handbook for Senior Clubs*. (Japan Federation of Senior Citizens' Clubs Inc. 2021).

15   Saito, Toru. 2016. The "Aging Problem" of "Elderly Club" Emerges (Fujou suru "Rojin Kurabu" no "Koreisha Mondai") 12 May 2016. Yahoo Japan. Available online: https://news.yahoo.co.jp/byline/torusaito/20160512-00057597 (accessed on 11 October 2021). (Saito 2016). Statistical Office, the Cabinet Office of Japan. 2020. *White Paper on the Ageing Society in 2019.* Figure 2-2-1.

16   Hearing from the staff of Japan Federation of Senior Citizens' Clubs Inc. on 10 August 2021.

17   Japan Federation of Senior Citizens' Clubs Inc. (2015). *National Survey Report on Senior Clubs in 2015.* There were 2215 Senior Clubs throughout Japan that responded with an 84.8% response rate.

18   Source from Japan Federation of Senior Citizens' Clubs. The national government's method of funding senior clubs changed in 1999. From 2000, the prefectural and local municipal governments submit their allocations to the nation government which in turn funds senior clubs. Although the allocation of funding has changed, the size of the financial support to senior clubs has not changed much.

19   This is a simplified golf-like sports activity which was created for older adults in 1982.

20   Ministry of Health and Labor. 2018. *Administrative Report on Welfare Measures in 2018.* (Ministry of Health and Labor 2018).

21   Personal communication with the staff of the National Association of Welfare Centers for the Elderly. 28 April 2021.

22   Department of Senior Support, Urayasu City. 2006. *Report on Facilities and Building Maintenance Plan for the Promotion of the Health and Well-being of Seniors*, (*Urayasu-shi Ikigai Kenkozukuri Shisetsu Seibi Kousou Sakutei Houkokusho*). p. 10. (Department of Senior Support 2006).

23    Nishi Nihon Shinbun (19 September 2021). Rapid Decline in Places of Origin: Senior Clubs are Shrinking even in Super-Aging Japan. ('*Hassho no chi' demo gekigen: Choukoureika shakai nanoni roujinkurabu ga hosoru wake*) (Nishi Nihon Shinbun 2021). This newspaper article points to the increasing number of older people working, the impact of the corona virus and the diversification of values. It suggests that traditional senior clubs including women's associations may not be meeting the needs of senior today.

24    Fuwa, Kazuhiko. 2001. Lifelong education in Japan, a highly school-centered society: Educational opportunities and practical educational activities for adults. *International Journal of Lifelong Education* 20: 127–36. (Fuwa 2001).

25    Ogden, Anthony C. 2010. A Brief Overview of Lifelong Learning in Japan. *The Language Teacher* 6: 5–13. (Ogden 2010).

26    Faure, Edgar, Felipe Herrera, Abdul-Razzak Kaddoura, Henri Lopes, Arthur V. Petrovsky, Majid Rahnema, and Frederick Champion Ward. 1972. *Learning to be: The World of Education Today and Tomorrow.* Paris: UNESCO. (Faure et al. 1972).

27    In 1984, the "Integrated Ikigai Promotion Program was established and by 1989, Choju Gakuen (Long Life Academy) was established.

28    Takahashi, I. 1999. *Meanings and Issues of Senior Education*. Sendai: Sendai Shirayuri Women's University, NII-Electronics Library Services, pp. 97–103 (Takahashi 1999). See also Hori, Shigeo. 2010. A Study on the Change of Function of a Senior College: A Comparative Study in Nishinomiya City Senior College in 10 Years (Koreisha Daigaku no Kinou no Henka ni kansuru Chosakenkyu). *Ronen Shakaigaku* 32: 338–47 (Hori 2010).

29    Kinoshita, Yasuhito. 2021. The COVID-19 Impact on Learning Programs for the Senior in Japan with a case of Rikkyo Second Stage College—University-Affiliated Program. Aging People and Society Symposium, UNESCO Turkish National Commission for UNESCO, Turkey, March 22 (online). (Kinoshita 2021).

30    Setagaya Senior College was established in 1977 and helped shape the programs in the other 23 wards of the Greater Tokyo Metropolitan area. See: Miura, Fumio. 1996. *Challenging to Study in Later Years at Setagaya Senior College*. Kyoto: Minerva Publishing Co. (Miura 1996).

31    Kinoshita, Yasuhito. 2018. *Senia Manabi no Gunsou: Teinengo Lifusutairuno Sousutsu (The Coming Wave of Senior Learners)*. Tokyo: Kobundo Publishing Co. (Kinoshita 2018).

32    List of Extension Courses for Seniors (University Open College). Available online: https://goodlifesenior.com/wp/news/7873 (accessed on 3 October2021).

33    https://www.nenshuu.net/prefecture/shotoku/in_shotoku_city.php (accessed on 15 August 2021).

34    http://area-info.jpn.org/KS02002All.html (accessed on 3 November 2021).

35    Statistic Office of Urayasu-City. 2021. Annual Report of Urayasu City in 2020. (Statistic Office of Urayasu City 2021).

36    The Association of Senior Clubs in Urayasu is the coordinating agency for 50 Senior Clubs in Urayasu City.

37    The Urayasu City history information obtained from the Bureau of General Affairs of Urayasu City.

38    White Pater on the Aging Society 2019, and Association of Urayasu Senior Clubs in 2020. (Bureau of Statistics, Cabinet Office of Japan 2020; Association of Urayasu Senior Clubs 2020).

39    Admission Handbook for 2020, Urayasu Citizen College (2019b). Urayasu City (2019). Also, Urayasu Citizen College (2019c). *Annual Report of 2019.* Urayasu City (2020), Japan.

40    Data is from the Bureau of General Affairs, Urayasu City.

41    Available online: https://en.wikipedia.org/wiki/Corporation for National and Community Service (accessed on 25 October 2021); See also Available online: https://www.americorps.gov/serve/fit-finder/americorps-seniors-senior-companion-program (accessed on 25 October 2021).

42    Available online: www.roadscholar.org (accessed on 10 August 2021).

43    Personal communications with Director of the Senior Clubs Program. City and County of Honolulu (June 2021).

44    Personal communications with Director of the Senior Clubs Program. City and County of Honolulu (June 2021).

45    Available online: https://oahuseniorsoftball.org/ (accessed on 10 November 2021).

46    Available online: https://www.hawaii.edu/diversity/seed-programs/na-kupuna-program/ (accessed 5 August 2021).

47    Available online: https://osher.socialsciences.hawaii.edu/ (accessed on 7 October 2021).

48    Since 1886 when compulsory education began, the literacy rate in Japan reached nearly 100% by 1945. Today, there are almost no illiterate older adults in Japan. Saito, Toru. 2012. Literacy rate and historical trend: in the case of Japan. In *Annuls of the Kokusai-Kyoryoku.* Hiroshima: Hiroshima University, International Educational Development Center, vol. 1, no. 1, pp. 51–62. (Saito 2012).

49    End of Life Council Association. 12 Universities That Senior Citizens Aged 50 and 60 Can Attend! What Are the Benefits of Going to College for Seniors? (50–60 Sai no Koureisha ga Kayoeru Daigaku 12 sen! Senia Daigaku e Kayou Merito Towa?). Available online: https://www.enjoy-mylife.net/others/university-merits (accessed on 25 August 2021). (End of Life Council Association 2021).

50    White, Sarah. 2021. 20 Places to Educate Yourself Online for Free. Available online: https://www.lifehack.org/articles/productivity/20-places-educate-yourself-online-for-free.html (accessed on 26 September 2021). (White 2021).

51   Available online: www.lifelongsociety.org (accessed o 11 September 2021). October 1 has been designated the Shogai Gen'eki Day in Japan.

52   Both public and private sectors, the mandatory retirement age will be extended from 2023 by Law. It will be extended from 60 to 65 gradually by the year in 2031.

53   Hayashida, Cullen. 2020. Japan's Depopulation and Workforce Shortage Crisis: An Overview of 11 Strategies and a Paradigm for a Macro-Research Agenda NTA2020 Hayashida. Available online: https://ntaccounts.org/web/nta/show/Documents/NTA2020%20Hayashida (accessed on 17 November 2021). (Hayashida 2020).

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
