# Peer review of "The Past, Present and Future Direction of Government-Supported Active Aging Initiatives in Japan: A Work in Progress"

_socsci, doi:10.3390/socsci11020065_

Round 1
Reviewer 1 Report
The article “The Past, Present and Future Direction of Government Sup-1 ported Active Aging Initiatives in Japan” addressed a novel and timely topic (two programmatic trends in Japan, one with Senior Centers/Senior Clubs and the other with Senior Colleges).
The paper, however, is more of a research report than a full article, as it is lacking a theoretical framing, hypotheses, research questions or objectives, connection between the theoretical framing and the results, and a discussion of the implications of the findings for international scholarship.
Author Response
November 28, 2021
Dear Reviewer 1:
First of all, we would like to thank you for your critique and suggestions to improve the paper. They were some consistency in both reviews that were helpful. Given the degree of overlap between both, we would like to provide an updated analytical framework or paradigm in the introductory section.
Reviewer #1 states that "the paper is lacking a theoretical framing, hypotheses, research questions or objectives, connection between the theoretical framing and the results and the discussions of the implications of the findings for international scholarship." Please see the revised manuscript.
Reviewer 2 Report
The author reviewed the active aging policy since the 60s in Japan with a focus on the historical development of Senior Clubs, Welfare Centers for the Elderly, and Senior Colleges, and a case report from a city. The authors also compare such policies with those of the U.S. The author attempted to bring out the difference and the characteristics of aging aging policy which are influenced by the governments’ attitudes through cross national comparisons. The reviewer does not have any doubts about the facts and historical background in the manuscript, especially regarding Japan. However, the reviewer does point out some problems, which should be improved.
The manuscript is not sufficiently analytical. The author took up Urayasu City as a case study in Japan, but the city is a special case for Japanese municipalities in terms of its aging and financial situation, as pointed out by the author. Therefore, it is necessary to show whether the case is sufficiently general.
The author discusses the Japanese case, the general situation in the US, and the case in Hawaii. However, only a few comparisons regarding these are made. It is necessary to clearly state the perspective of the comparison and to conduct analysis.
The author should have more discussion. Japan has been providing policy support for active aging initiatives and has been prevailed in the community. As the author pointed out, the participation rate of senior clubs has been declining and will probably continue to decline in the future. For the benefit of readers who are not familiar with the situation and system in Japan, the author should state the factors behind the decline in regional senior activities supported by national and regional governments. It is also necessary to summarize and evaluate the effects of such active aging policy. In addition, it would be helpful to the readers if the author discusses what will happen to active ageing initiatives in the future if the local activities continue to decline, whether there are alternative measures, and what the impact of the decline will be.
Author Response
November 28, 2021
Dear Reviewer: 2
Thank you for your critique of the manuscript. What follows are our comments to address issues that you raised and our attempt at addressing them in the text of the manuscript.
- The manuscript is not sufficiently analytical: The manuscript has been revised to address this criticism that it was insufficiently analytical. This has been addressed with another section at the beginning of the revised manuscript. The theoretical issue relates to the sustainability of societies given depopulation trends that are affecting scores of societies today and more in the future. We've pointed to a number of trends and issues such as population aging, intergenerational conflicts, and the medical and institutional bias of current gerontological theories to address the issue of societal sustainability. It is within this context that active aging is referred to as a relatively new and emerging field in international gerontology. The question is to what extent do policies and strategies based on this perspective represents an effective strategy for societal sustainability. Given its status as a super-aging society and a pioneer in active aging program development, what happens in Japan needs to be monitored and documented. What is reported is descriptive and reflective of the changes that it has undergone since the 1960s. It is evident that Japan's policies and practices are adapting in its conceptualization of active aging and this report attempts to point to ways that the changing perspective is evolving.
- The Case of Urayasu City as a Case Study: The reviewers are correct. The case of Urayasu city is not typical of most municipalities in Japan given that its social welfare services are very well developed. This is a point that the authors noted in the report. The Urayasu case study, however, provided concrete documentation of the implementation of existing national and local policies in support of active aging. It may be viewed as an ideal-typical example of what other municipalities would or could do if they had the resources.
- The general situation of the US and the Hawaii case study: Similarly, the comparison with the U.S. and Hawaii situation was a heuristic means of noting how unusual Japan has been in terms of its rate of program development, funding, and implementation. The US and Hawaii comparison was used as a benchmark. Any other national comparison could have served the same purpose. If the reviewers prefer, this section of the report could be omitted.
- Factors related to the decline in regional senior activities: Possible factors related to the decline have been suggested in the manuscript. Among those noted were changes in the role of older adults in the nuclearization of the family structure, urbanization, economic growth, and increased financial resources with pensions and health plans, and the like.
- Summarize and evaluate the effects of active aging policy: This paper is an early attempt at bringing attention to the rapid changes that are occurring in Japan. The paper has attempted to summarize some of the changes (eg. decline in participation, variation in urban vs rural activity levels, the emergence of senior college programs, etc.) however, this entire is a moving target and will require better analytic tools and theoretical modeling in the future. Hopefully, other countries that are experiencing population decline will find reasons to collaborate with future studies in this area.
- What happens if the active aging initiatives does not work? Are there alternative measures? What is the impact of the decline? We have made reference to a preliminary report by Hayashida that the 2020 International Generational Economy conference in Honolulu, Hawaii. In that presentation, Hayashida noted that there are a number of measures that are under consideration to moderate or reverse Japan's depopulation trend. They include the use of technology, the importation of foreign workers, increase birthrate, postponement of retirement age, development of age-friendly communities, deurbanization measures, increase women in the workforce, and expansion of childcare support infrastructure. At this time, we do not know if active aging is a viable national strategy to address the long-term sustainability of a society. It is probably necessary but insufficient. A long-term solution will probably require the combined impact of a number of strategies. Which and what combination is a question that goes beyond the scope of this paper. It is hoped that this report can be part of a larger multi-national research and policy discussion.
Round 2
Reviewer 2 Report
The reviewer read the author's responses, but I could not understand the author's intentions and the policy of the revision. The authors made a major revision only to the introduction, which makes the thesis of the paper even more unclear. The lack of proper citation in this section is also a problem. The reviewer expects authors to be more honest in their revisions.
Author Response
Reviewer 1 stated "Reviewer #1 states that "the paper is lacking a theoretical framing, hypotheses, research questions or objectives, connection between the theoretical framing and the results and the discussions of the implications of the findings for international scholarship."
In revising this manuscript we wish to make it clear that is this more of a trend analysis of a public policy perspective related to active aging. While we make reference to the theoretical grounding of the concept of active aging in the literature by the World Health Organization, the International Council on Active Aging, and a more recent article by Dr. Leng Leng Thang, we are not proposing a hypothesis and/or research findings based on primary data. What we are principally addressing is the evolution of the active aging policies and programs that are changing as a result of the socio-demographic changes in Japan itself. Japan is now about to make further refinements to its active aging strategy by it future focus on employment. Thus, our efforts are to document the evolution of the active aging policy in Japan as a work in progress.
Reviewer #2 stated "The authors made a major revision only to the introduction, which makes the thesis of the paper even more unclear. The lack of proper citation in this section is also a problem. The reviewer expects authors to be more honest in their revisions.
We apologize for any misunderstandings. We have again substantially revised the introductory section and inserted a summary discussion section before the shorter conclusion. We have also included added references where they were necessary.
Round 3
Reviewer 2 Report
I do not have any further comments on the manuscript.